# Does innovation policy drive rural revitalization in China? - An empirical study based on Chinese rural areas

Zhige Ying[1]*, Cheng Zhang[1], Hande Chen[1], Zhiyuan Cheng[2], Zhe Wu[3]

**1** School of Innovation and Entrepreneurship, Hubei University of Technology, Wuhan, Hubei, China, **2** Industrial Design College, Hubei University of Technology, Wuhan, Hubei, China, **3** College of Energy and Power, Naval University of Engineering, Wuhan, Hubei, China

\* yzg@163.com

**Data availability statement:** The data supporting the findings of this study are available in Dryad (DOI: https://doi.org/10.5061/dryad.qv9s4mwr5).

**Funding:** This work was supported by the Natural Science Foundation of Hubei Province (grant number 2017AAA108).

**Competing interests:** The authors have declared that no competing interests exist.

## Abstract

Innovation serves not only as the cornerstone of rural revitalization but also as the fundamental driving force behind agricultural and rural modernization in China. Since 2008, China has initiated the implementation of innovative city pilot projects, gradually expanding their scope. This study, based on mechanism analysis, employs panel data from 280 cities and local rural areas spanning from 2004 to 2018 to empirically examine the impact of pilot policies on local rural revitalization levels using methods such as the difference-in-differences model. The research findings indicate that innovative city pilot policies significantly enhance rural revitalization levels. Heterogeneity analysis reveals that the effect of innovation policies is pronounced in cities with high entrepreneurial activity but shows no significant correlation with city level, or science and education resources. Furthermore, the study finds that innovative city pilot policies contribute to strengthening local rural communication levels, cultural and educational development, promoting talent aggregation, and indirectly facilitating other policies supporting rural areas, thereby positively influencing rural development. Additionally, the aggregation effect of talents and the incentive effect on business investment are crucial reasons for enhancing urban innovation levels. The conclusions drawn in this study provide policy insights for fully unleashing the rural revitalization effects of innovation-driven policies, thus promoting high-quality economic development in China and enhancing the international community's understanding of Chinese policies and rural areas.

## Introduction

The greatest imbalance in China's development lies in the disparity between urban and rural areas. Although China has achieved the goal of comprehensive well-being, the income gap between urban and rural residents is expected to remain at a high level for the long term [1]. In particular, the decline in birth rates since the 1980s has significantly increased the aging population in rural areas. This, coupled with the outflow of labor, has further weakened the development of rural areas and small towns [2]. Rural education and healthcare have been key issues for the Chinese government to address since the 21st century. Under market economy

conditions, education exhibits clear externalities, particularly in rural areas where the marketization of education is low and competition is insufficient, leading to inadequate supply of educational services [3]. Healthcare services often display characteristics of "market failure." On one hand, due to low income levels, many rural residents are unable to afford high medical costs. On the other hand, there is a severe "supply-side" issue, as rural areas lack sufficient hospitals and clinics [4].

As China's economy and society enter a new era, especially after the contradiction between the growing demand for a better life and the unbalanced and insufficient development became the primary contradiction in the new era's construction, the importance and value of rural development have been elevated, creating conditions for rural revitalization. To address the severe issues faced by rural areas, the Chinese government proposed the Rural Revitalization Strategy in 2017 to promote integrated urban-rural development. This proposal marks a shift in urban-rural relations and reflects a significant change in ideology [5]. Rural revitalization involves not only economic development, but also social, cultural, and ecological aspects, playing a crucial role in building a more balanced, healthy, and sustainable societal system [6, 7]. The comprehensive advancement of rural revitalization requires the collective effort of various stakeholders, including the government and enterprises [8]. However, empirical studies on rural revitalization remain relatively scarce, with existing research mainly focusing on specific provinces, cities, or poverty-stricken counties, and lacking comprehensive assessments at the level of prefecture-level cities nationwide [9].

Innovation in agriculture, rural areas, and among farmers is not only the foundation of China's stability and well-being but is also critical for other developing countries. By building a strong and efficient innovation system, rural areas can overcome long-standing challenges such as poverty, poor land management, inadequate infrastructure, and crime, contributing to global sustainable development [10]. Matthews [11] argued that in West Africa, supporting indigenous innovation processes rather than focusing on initiating change can be a more prudent approach to promoting rural development. Using India as an example, the traditional innovation pathway of scaling-up should leverage the dynamics of peri-urban areas, including building new alliances to renegotiate governance structures across the rural-urban continuum, reframing the discussions on urban sustainability, and reconfiguring interactions between socio-technical and social-ecological systems [12]. In the Middle East, a thorough reform of governance procedures and institutional arrangements can reduce ethnic, religious, and political conflicts to enhance rural innovation and entrepreneurship [13].

Innovation-driven rural revitalization, as a unique pathway for rural development, has garnered widespread attention in academia. Studies have shown that government support, market demand, and innovation investment are key factors in implementing this strategy, but the homogenization and low-end nature of innovation outcomes may limit their effectiveness [14]. Scholars have also summarized the practical experiences of key innovation entities such as universities, research institutions, agribusinesses, and talent, as well as the roles of innovation platforms like agricultural parks and research bases [15,16]. Jiang [17] points out that agricultural mechanization innovation not only improves labor productivity but also reshapes rural production relations, lifestyles, and governance systems, thus providing momentum for rural revitalization. In recent years, rural areas have been facing the challenge of population decline. To effectively address this issue, the formulation of innovative policies needs to be tailored to local conditions, taking into full account the actual development status of the region. Only in this way can these policies be effectively implemented and further developed in areas experiencing rural population decline [18]. In the early 20th century, researchers suggested that innovations in information technology would significantly enhance the development levels of rural areas [19]. Although previous research has explored

the impact of innovation on rural areas and society as a whole, there remain gaps in empirical analysis and research on spatial effects. Furthermore, existing literature has not yet integrated innovation pathways with their mechanisms at the macro-policy level, nor has it thoroughly discussed the comprehensive impact of innovation on rural revitalization.

## Mechanism analysis

The mechanism driving innovation-based rural revitalization stems from multiple dimensions, as suggested by existing research. First, innovative technologies and management models optimize the allocation of rural resources, enhancing agricultural productivity [20]. Applications in areas such as information technology, smart agriculture, and environmental sustainability not only improve production methods but also increase the sustainability and resilience of agriculture. Second, innovation fosters the diversification of rural industries, particularly the integration of agriculture, tourism, and culture, thereby driving the transformation and upgrading of the rural economy [21]. This industrial synergy not only strengthens the economic vitality of rural areas but also injects new growth drivers into rural economies.

Innovation also addresses rural "hollowing-out" by optimizing business models and infrastructure, attracting talent and capital inflows, which in turn stimulates endogenous growth in rural areas. Specifically, the path to innovation-driven rural industrial revitalization can be divided into three stages: the primary stage, focusing on R&D, original innovation, and infrastructure development; the intermediate stage, emphasizing the commercialization and application of scientific results to promote the dissemination of technology; and the advanced stage, where technological promotion and market expansion become the focus, driving the implementation of technological and business model innovations. The innovation-driven mechanism primarily includes factor-driven, demand-driven, and competition-driven forces. Factor-driven innovation enhances productivity by introducing technological elements, demand-driven innovation stimulates product and service development through market needs, and competition-driven innovation incentivizes firms and local governments to innovate, pushing industries toward higher technological content and greater value-added. Ultimately, innovation not only enhances the comprehensive competitiveness of rural industries but also lays the foundation for the long-term revitalization of rural areas.

Currently, China's economy is undergoing a critical transition from high-speed growth to high-quality development [22]. As a gradual reform policy for implementing the innovation-driven development strategy, the national innovation-driven urban pilot policy has gone through six rounds of approval for pilot cities, covering multiple national innovation pilot cities (districts) in five provinces. This policy framework represents the core content of China's innovation-driven policies. The primary goal of the policy is to build innovation-driven cities, enhance urban autonomous innovation capacity, and achieve innovation-driven development. The policy encompasses areas such as strengthening scientific research and development, offering tax incentives, fostering talent aggregation, and improving government services. It involves multiple stakeholders, including government, industry, academia, research, and finance, forming a relatively systematic policy framework to promote innovation-driven development.

The founding father of classical economics, Adam Smith [23], proposed the concept of "natural order." He argued that villages preceded cities and that cities emerged from rural development; the wealth of a nation largely depends on factors such as historical geography, institutional frameworks, and cultural aspects influencing urban and rural development. Influenced by this idea, German economic geographer Johann Heinrich von Thünen [24] viewed cities and rural areas as an integrated whole in his "Isolated State" model, exploring

the spatial distribution patterns of various industries between urban and rural areas. Western urban studies can be traced back to the city-state theories of ancient Greece, where the Greeks linked city-state construction with politics and civilization [25,26]. Plato's concept of Utopia represents one of the earliest ideal governance models in human history [27]. Advocates of Utopia strongly endorsed the integration of cities and rural areas and further emphasized their coordinated design. Advocates of Utopia strongly endorsed the integration of cities and rural areas and further emphasized their coordinated design [28]. Some scholars point out that both developed and developing countries will realize a new type of urban-rural integration characterized by complementarity, symbiosis, and shared prosperity [29–31]. Particularly under the influence of the digital economy, the process of urban-rural integration will accelerate further [32]. Scholars also argue that globalization and urbanization are continually transforming and reshaping urban-rural relations, with the evolution of global urban-rural relationships expected to follow an orderly progression from urban-rural division and confrontation to coordination and integration [33].

In the process of urban-rural integration, pilot policies for innovative cities, with a core focus on technological innovation, can leverage knowledge and technology spillover effects to bring innovative outcomes to rural areas [34]. For instance, the application of emerging technologies such as the Internet+, big data, and artificial intelligence can promote agricultural modernization, improving both agricultural productivity and quality [35]. As the industrial and innovation chains in innovative cities become more refined, the cities' advantages in resources, technology, and talent can radiate to surrounding rural areas, stimulating rural economic development [36,37]. Innovation policies are not limited to the economic sphere but also extend to the improvement of social governance and public services [38]. For example, policy innovations in areas such as smart cities and digital rural construction [39] can enhance governance efficiency in rural areas, improve the quality of public services, and improve the living conditions of rural residents.

The Innovative Cities Pilot Policy promotes rural revitalization through a multidimensional mechanism covering policy innovation, technological innovation, organizational innovation, business innovation, financial service innovation and talent development. Policy innovation stabilizes the land contract relationship, improves the land system and increases national inputs to help modernize agriculture; technological innovation improves the efficiency and quality of agriculture through Internet+, big data and other technologies; organizational innovation solves the contradiction between small farmers and the big market through rural cooperatives and other new business subjects; industrial innovation activates the rural economy through the development of leisure agriculture, e-commerce and integration of one, two and three industries; financial service innovation Optimization of rural financial policies, unblocking the flow of capital, providing capital support for rural development; talent development through the training and introduction of high-quality talents, to provide intellectual support for rural revitalization. In addition, through the spillover effects of technology, resources, capital and talent, the policy promotes integrated urban-rural development, leads to synergistic regional progress, comprehensively improves the economic and social level of the countryside, and injects sustained impetus into the realization of socialist modernization,The mechanism of action is shown in Fig 1.

Innovation is a critical driver of economic growth, and rural revitalization is a strategic initiative to address China's urban-rural imbalance. Innovation, as a cornerstone of rural revitalization, enhances productivity, improves living standards, and fosters sustainable rural development. Among China's measures to boost innovation, the innovative city pilot policy plays a pivotal role in advancing urban innovation while supporting national innovation goals. This study aims to answer a central question: can innovation policies effectively promote rural

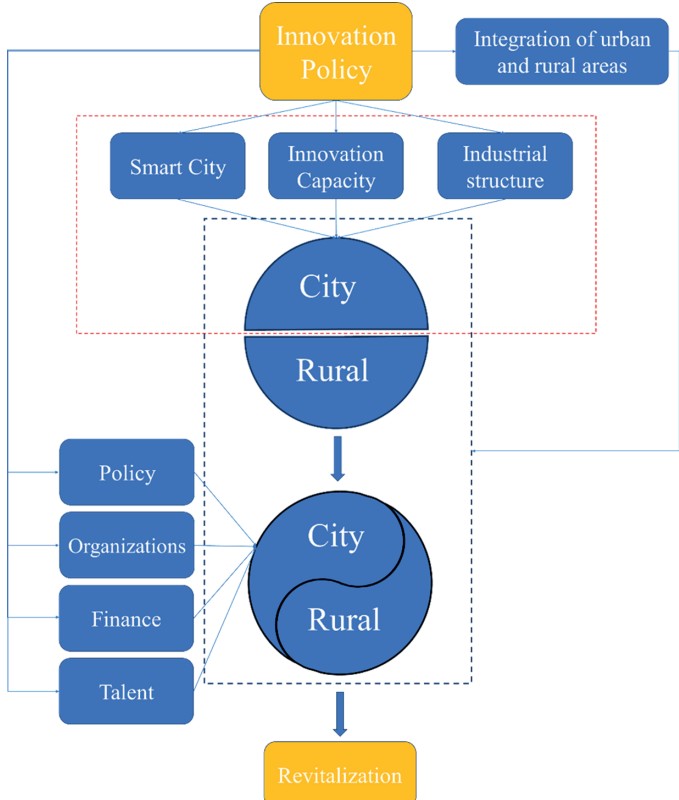

**Fig 1. Innovative policies to drive rural revitalization.**

revitalization? Addressing this issue not only contributes to the debate on the effectiveness of government innovation policies but also provides policymakers with actionable insights for designing targeted support measures.

## Data description and model design

### Multiple time periods difference-in-differences (DID) benchmark regression model

Given that China has been conducting innovative city pilot projects since 2008, the National Innovation-Driven City Pilot Policy of China can be viewed as an exogenous policy shock to the Rural Revitalization Index, making it akin to a quasi-natural experiment of implementing innovation policies. Considering that the innovative city pilot policy gradually expands the scope of pilot cities, in order to scientifically assess the impact of pilot policies on rural development, we constructed the following regression model as Eq (1):

$$\text{Rural}_{\text{index}_{i,t}} = \alpha + \beta\,\text{Inno}_{\text{Policy}_{i,t}} + \gamma\,\text{Control}_{\text{Var}_{i,t}} + \text{CityFE} + \text{YearFE} + \varepsilon_{i,t} \tag{1}$$

Where $Rural_{index_{i,t}}$ represents the Rural Revitalization Index, $Inno_{Policy_{i,t}}$ denotes the Innovative City Policy, and $Control_{Var_{i,t}}$ stands for multiple control variables. Due to variations in the years and cities of the innovative city pilot program, *CityFE* and *YearFE* are respectively

introduced as fixed effects. Here, $\alpha$ describes the average difference before and after the policy implementation, while $\varepsilon$ represents the random disturbance.

## Rural revitalization index and variable design

(1) The dependent variable $Rural_{index_{i,t}}$ (rural revitalization index)

In 2018, the Chinese government proposed the "Rural Revitalization Strategic Plan (2018–2022)", which explicitly outlines the requirement to promote rural industry, talent, culture, ecology, and organization revitalization in a scientific and orderly manner, according to the overall requirements of "industrial prosperity, ecological livability, civilized rural customs, effective governance, and affluent living [40]." In the implementation of rural revitalization, industrial prosperity is the focus, ecological livability is crucial, civilized rural customs are guaranteed, effective governance is the foundation, and affluent living is fundamental. Based on the elucidation of the connotation of rural revitalization above, and following principles of scientificity, feasibility, measurability, and data accessibility, this paper draws on the research of Xu Xue and Wang [41] to calculate the comprehensive index of rural revitalization and the indices of five subsystems using the entropy method. The specific indicators are shown in Table 1, The weights of each indicator are evenly distributed , the calculation methods are provided in Eqs (2), (3), (4), (5), and (6).

$$X'_{i,j} = \left[ \frac{\max\left(X_{1j}, X_{2j}, \cdots, X_{nj}\right) - X_{i,j}}{\max\left(X_{1j}, X_{2j}, \cdots, X_{nj}\right) - \min\left(X_{1j}, X_{2j}, \cdots, X_{nj}\right)} \right] + 0.01 \tag{2}$$

$$P_{ij} = \frac{X'_{ij}}{\sum_{i=1}^{n} X'_{ij}} \tag{3}$$

$$P_{ij} = e_j = \qquad 1/\ln(n) \sum_{i=1}^{n} P_{ij} \ln(P) \tag{4}$$

$$w_j = \frac{1 - e_j}{\sum_{j=1}^{m} 1 - e_j} \tag{5}$$

$$P_{ij} = S_i = \sum_{j=1}^{m} w_j X'_{i,j} / \% \tag{6}$$

Where $X'_{i,j}$ represents the value of the j-th indicator for the i-th city after processing, with i = 1, 2, ..., n and j = 1, 2, ..., m. $P_{ij}$ represents the proportion of the j-th indicator in the total for the i-th city, $e_j$ represents the entropy value of the j-th indicator, $w_j$ represents the weight of the j-th indicator among all indicators, and $S_i$ represents the comprehensive score of each province. Taking the logarithm of this value gives the core dependent variable $rural\_index$

(2) The core explanatory variable $Inno\_Policy_{i,t}$ (innovation policy)

This paper regards the National Innovation-Driven City Pilot Policy as a quasi-natural experiment. China's innovative cities pilot is divided into three batches, which were established by the National Development and Reform Commission (NDRC) and the Ministry of Science and Technology (MOST) of China, and gradually implemented over the years. The specific timelines are shown in Table 2 .To provide a clearer description of the impact of innovation policy, this study utilizes the product of city type and the implementation of innovation policy to characterize the policy treatment effect of innovation-driven policy ($Inno\_Policy_{i,t}$). The National Innovation Cities are designated as the experimental group, denoted by the Group value of 1, while non-national innovation pilot cities are designated as 0, serving as the control group. If the policy has been implemented in a certain year, the time

**Table 1. Rural revitalization evaluation system.**

| Assessment System | Dimension | Norm |
|---|---|---|
| Industrial prosperity | Agricultural production capacity base | Per capita total agricultural machinery power (kilowatts) |
| | | Comprehensive grain production capacity (ten thousand tons) |
| | Industry integration level | Main business income of agricultural product processing enterprises above the designated size (hundred million yuan) |
| Ecological livability | Green agricultural development | Pesticide and fertilizer application amount (ten thousand tons) |
| | | Comprehensive utilization rate of livestock and poultry manure |
| | Rural living environment governance | Percentage of administrative villages treating domestic wastewater (%) |
| | | Percentage of administrative villages treating domestic waste (%) |
| | | Sanitary toilet coverage rate (%) |
| | Rural ecological protection | Rural greening rate (%) |
| Civilized rural customs | Educational level of farmers | Percentage of full-time teachers in rural compulsory education schools with a bachelor's degree or higher (%) |
| | | Average years of education of rural residents (years) |
| | Traditional culture dissemination or Dissemination of traditional culture | Cable television coverage rate (%) |
| | | Percentage of administrative villages with internet broadband services (%) |
| Effective governance | Governance capacity or Governance ability | Percentage of village heads and party secretaries holding both positions (%) |
| | | Percentage of administrative villages with a developed village plan (%) |
| | Governance measures | Percentage of administrative villages that have carried out village remediation (%) |
| Affluent living | Farmer income level | Per capita net income of farmers (yuan) |
| | | Per capita income growth rate of farmers (%) |
| | Consumption structure of farmers | Engel's coefficient of rural residents (%) |
| | Living conditions of farmers | Number of cars per 100 households (vehicles) |

variable Post is set to 1; otherwise, it is set to 0. This setup allows for a significant distinction between cities before and after policy implementation.

(3) Control variables *Control_Var*

In empirical research, control variables are set to ensure the accuracy and reliability of the study results. By controlling for these factors in the research design, researchers can more accurately analyze and interpret the relationship between the core explanatory variables and the dependent variables. Therefore, this study sets the following control variables:

1. Local GDP (*lnagdp*): By introducing local GDP as a control variable, other factors related to rural development can be better controlled, allowing for a more precise assessment of the independent impact of innovation on rural development, ensuring the accuracy of causality [42]. Moreover, observing the relationship between innovation and rural development during the empirical process as causal rather than due to other factors helps establish a more credible causal relationship.

2. Rural Per Capita Consumption (*rural_consume*): Rural per capita consumption is an important indicator of the living standards of rural residents. Higher per capita consumption typically indicates that rural residents have more disposable income to spend

**Table 2. Time and batches of innovation pilot city designation.**

| Batch | City | Approved by | Year |
|---|---|---|---|
| 1 | Chengdu, Changsha, Dalian, Guangzhou, Hefei, Hangzhou, Jinan, Nanjing, Qingdao, Shenyang, Suzhou, Xi'an, Wuxi, Yantai, Xiamen | NDRC | 2010 |
| 1 | Baotou, Beijing, Changsha, Chengdu, Chongqing, Guangzhou, Harbin, Hefei, Jiaxing, Jinan, Lanzhou, Luoyang, Nanjing, Ningbo, Shanghai, Tangshan, Tianjin, Wuhan, Xiamen, Xi'an | MOST | 2010 |
| 2 | Baoji, Changchun, Changzhou, Dalian, Fuzhou, Guiyang, Haikou, Jingdezhen, Kunming, Luojiang, Nanchang, Nanning, Shenyang, Shihezi, Shijiazhuang, Taiyuan, Xining, Yinchuan | MOST | 2010 |
| 3 | Hohhot, Lianyungang, Taihuangdao, Zhenjiang | MOST | 2011 |
| 3 | Nantong, Urumqi, Zhengzhou | MOST | 2012 |
| 3 | Huzhou, Jining, Nanyang, Pingxiang, Taizhou, Xiangyang, Yancheng, Yangzhou, Yichang | MOST | 2013 |
| 3 | Dongguan, Dongying, Foshan, Hengyang, Hanzhong, Jilin, Jinhua, Lhasa, Longyan, Ma'anshan, Quanzhou, Shaoxing, Weifang, Xuzhou, Yuxi, Zhuzhou, Wuhu | MOST | 2018 |

on consumption over a certain period, enjoying more material goods and services. Additionally, higher per capita consumption may be associated with better social welfare and public services, including better healthcare, education, infrastructure, etc. [43], thereby improving residents' quality of life. The level of living standards in an area may influence the demand for and acceptance of innovation in that area. Considering the differences in living standards, including rural per capita consumption as a control variable helps more accurately assess the independent impact of innovation on rural development.

3. Marketization Level (*Market*): Rural revitalization involves multiple aspects, including policy support, industrial structure, innovation capabilities, etc. Changes in the level of marketization may be influenced by various factors. Including it as a control variable helps eliminate potential interference from other factors on the impact of innovation-driven rural revitalization. Studies indicate that indicators such as the proportion of GDP created by the market sector, the level of market competition, the existence of market-oriented policies and regulations, and the degree of privatization can be used to measure the level of marketization [44]. Therefore, this study characterizes the degree of marketization by the ratio of government public budgets to GDP, where a smaller value indicates less government intervention in the market and a higher level of marketization.

4. Internet Penetration Rate (*Ainternet*): The internet penetration rate can affect the efficiency and effectiveness of rural technological innovation. Increasing the internet penetration rate benefits rural technological innovation subjects in utilizing internet platforms and tools for online learning, training, consulting, services, etc. [45], thereby enhancing the quality and level of rural technological innovation.

## Data source

This paper aims to explore the mechanism and path of innovation-driven rural industrial revitalization, in order to provide theoretical guidance and policy suggestions for the implementation of rural revitalization strategies. The data sources of this paper mainly include: China Rural Revitalization Survey (CRRS) conducted by the Rural Development

Institute of the Chinese Academy of Social Sciences. This survey is a comprehensive nation-wide survey covering multiple fields such as rural society, economy, culture, and ecology, providing rich first-hand data for analyzing the current situation and problems in rural areas. The remaining original data mainly come from "China Digital Rural Development Report," "China Rural Statistical Yearbook," "China Population and Employment Statistical Yearbook," "China Urban and Rural Construction Statistical Yearbook," "China Education Statistical Yearbook," "China Urban and Rural Statistical Yearbook," "China Social Statistical Yearbook," "China Civil Affairs Statistical Yearbook," "China Tertiary Industry Statistical Yearbook," and WIND data platform. In the process of data utilization, cities with severely missing data are excluded, and linear interpolation is used to fill in missing data for a small amount of missing data. Since China's "Rural Revitalization Strategic Plan" was first released in 2018, the data selected for this paper mainly covers the period from 2004 to 2018 in order to avoid conflating the impact of the Rural Revitalization strategy with that of the innovation-driven policy strategy.

## Empirical analysis

### Benchmark regression

The descriptive statistics of the main variables in this study are presented in Table 3. In the empirical analysis, the data is first subjected to Benchmark regression to analyze the impact of innovation policies on the rural revitalization index. The regression results are shown in Table 4 Benchmark Regression. In the table, (1) represents the regression results without adding control variables and without considering fixed effects, (2) represents the results considering fixed effects, (3) and (4) represent regressions with all control variables added, but (3) does not consider fixed effects. The regression results indicate that regardless of whether control variables and fixed effects are considered, the coefficient of the core explanatory variable $Inno_Ppolicy_{i,t}$ is positive and significant at the 1% level. As mentioned earlier, innovation policies play an important role in promoting economic, social, and environmental development in rural areas [46]. Innovation policies can stimulate endogenous dynamics in rural areas, promote optimization of rural industrial structure, enhance rural competitiveness and attractiveness. Innovation policies can also nurture rural talent resources, improve rural technological level and innovation capability, and enhance rural cultural vitality and social cohesion.

### Parallel trends test

The parallel trends test is a method used to evaluate the effectiveness of policy interventions. It is based on a crucial assumption that, in the absence of policy intervention, the trends in the outcome variables for the treatment group and the control group are the same. The purpose of the parallel trends test is to examine whether this assumption holds true, specifically whether

**Table 3. Descriptive statistics.**

| Variable | Definition | Obs | Mean | Std.Dev. | Min | Max |
|---|---|---|---|---|---|---|
| Rural_index | Rural Revitalization Index | 4170 | 28.96 | 10.27 | 4.93 | 72.97 |
| Inno_policy | Innovative City Pilot Policy | 4200 | 0.11 | 0.31 | 0 | 1 |
| Inagdp | Level of Economic Development | 4200 | 9.53 | 0.70 | 7.66 | 12.40 |
| Rural_ consume | Consumption level | 4170 | 5167.89 | 4291.06 | 223 | 253 |
| Market | The degree of Marketization | 4200 | 7.50 | 3.45 | 0.51 | 24.69 |
| Ainternet | Internet Penetration | 4200 | 14.38 | 16.3 | 0.07 | 198.6 |

**Table 4. Benchmark regression.**

| VARIABLES | (1) rural_index | (2) rural_index | (3) rural_index | (4) rural_index |
|---|---|---|---|---|
| Inno_Policy | 7.1451*** | 1.0334*** | 1.9232*** | 0.9835*** |
| | (14.5374) | (2.7434) | (3.7219) | (2.6322) |
| Observations | 4,170 | 4,170 | 4168 | 4168 |
| R-squared | 0.0483 | 0.7806 | 0.1853 | 0.7857 |
| CityFE | NO | YES | NO | YES |
| YearFE | NO | YES | NO | YES |
| Control_Var | NO | NO | YES | YES |
| Number of city | | 278 | | 272 |

there are differences in the changes of the Rural Revitalization Index before and after the policy implementation. If differences exist, it indicates that there were some structural differences between the treatment and control groups before the intervention. As a result, the estimation of the policy intervention effect may be biased. Therefore, this paper employs a multi-time point difference-in-differences (DID) model for testing. To ensure consistency between the treatment and control groups before and after the policy implementation, the relative time value is set as a dummy variable. The specific equation is as follows Eq (7):

$$\text{Rural }_{\text{index}_{i,t}} = \alpha + \alpha_1 \text{ Front }_{i,t} + \alpha_2 \text{ Front }_{i,t} + \alpha_3 \text{ Front } 1_{i,t} \tag{7}$$

$$+ \alpha_4 \text{ Occur }_{i,t} + \alpha_6 \text{ Afterwards } 1_{i,t} + \alpha_7 \text{ Afterwards }_{i,t} \tag{8}$$

$$+ \alpha_8 \text{ Afterwards } 3_{i,t} v + \alpha_9 \text{ Afterwards } 4_{i,t} + \alpha_{10} \text{ Afterwards } 5_{i,t} \tag{9}$$

$$+ \alpha_{11} \text{ Afterwards } 6_{i,t} + \beta \text{ Contro } + \text{ CityFE } + \text{ YearFE } + \varepsilon_{i,t} \tag{10}$$

The multi-time point difference-in-differences (DID) model has been extensively studied, and research indicates that different testing methods have varying effects on reflecting the robustness and accuracy of the model when researchers use the DID procedure. During the parallel trends test, it is advisable to select data that are exogenous and contextually relevant [47]. After observing the data, the first batch of innovative policies occurred in Shenzhen in 2008, reaching their peak scale in pilot cities in 2010. Moreover, most cities did not have samples from three years before the policy implementation. Therefore, this paper only examines the three years before and the six years after the policy implementation. The results of the test are shown in Fig 2. The results indicate that before the policy implementation, the regression coefficient of the relative time dummy variable is small and not significant. In the year of policy implementation, the coefficient significantly increases, experiences a slight decline in the following year, and then shows a trend of gradual increase year by year. This suggests that in the year immediately following the policy implementation, the pilot policy briefly generates rural revitalization effects but has not yet stabilized. One year after the implementation of the innovative policy, the coefficient of the impact of innovative city pilot policies becomes significantly positive and continues to rise, indicating that innovative policies can generate positive effects on promoting rural development. However, there is a certain lag effect.

## Endogeneity issues and robustness checks

**Endogeneity issues.** The implementation of innovation policies does not constitute a purely natural experiment, which raises the potential issue of selection bias. Therefore, this study adopts the methodology proposed by Derrien and Kecskes [48] and employs the

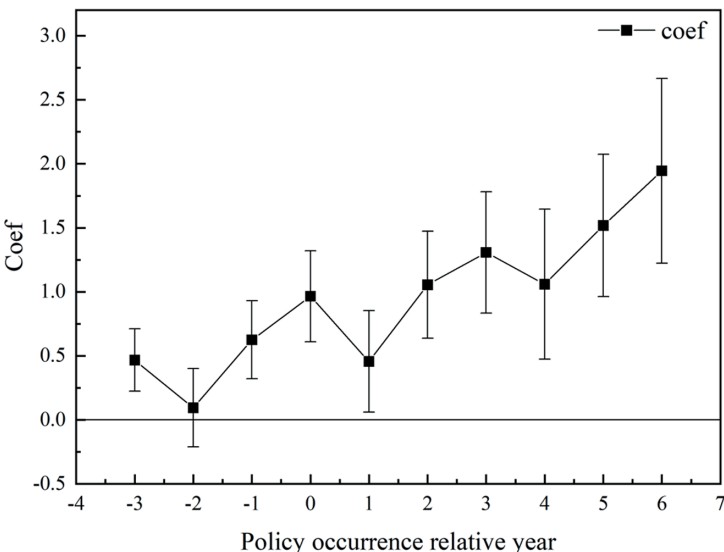

**Fig 2. Results of parallel trends test.**

Propensity Score Matching-Difference-in-Differences (PSM-DID) model for evaluation. This model integrates the propensity score matching (PSM) method with the difference-in-differences (DID) approach. Specifically, the PSM model is used to identify control units for treated units, while the DID model subsequently measures the causal impact of the policy intervention.

We selected local GDP, rural per capita consumption, degree of marketization, and internet penetration rate as matching variables and constructed a cross-sectional dataset. During the matching process, the nearest-neighbor matching method was applied to identify optimal control groups that share a common support with all innovative pilot cities, thereby generating a new dataset. By excluding observations that fall outside the common support, we refined the dataset to provide a more robust foundation for subsequent analysis. Additionally, we employed an iterative matching strategy. Under this approach, we matched city samples on an annual basis and then merged the matched data for all years into a longitudinal dataset suitable for regression analysis. This method allows us to capture the temporal evolution of city-level variables, providing a more comprehensive data perspective to support an in-depth understanding of the research question. To ensure the validity of the matching process, we conducted a balance test between the treatment and control groups across the matching variables. As shown in Table 5, the standardized mean differences of matching variables significantly decreased after matching. Fig 3 further illustrates that the standardized bias across these variables was reduced and maintained within the common support range.

The results indicate that, after matching, the treatment and control groups achieve high levels of balance across the matching variables. Consequently, we conclude that the matched sample satisfies the parallel trends assumption required for DID analysis. On this basis, we proceeded to evaluate the policy effects using the DID method. The final regression results of the PSM-DID model are presented in columns (1) and (2) of Table 6. The coefficients of the core explanatory variables are significantly positive, indicating that the implementation of innovation policies has had a promotive effect on rural revitalization to a certain extent.

**Table 5. Standardized deviations before and after matching.**

| Variable | Match | Mean | | %bias | %reduct |
|---|---|---|---|---|---|
| | | Treated | Control | | |
| *lnagdp* | U | 10.25 | 9.432 | 157.8 | 97.20 |
| | M | 10.24 | 10.26 | -4.500 | |
| *Rural Consume* | U | 9158 | 6720 | 61.40 | 82 |
| | M | 9050 | 8611 | 11 | |
| *Market* | U | 9.333 | 6.445 | 104 | 98.6 |
| | M | 9.275 | 9.233 | 1.500 | |
| *Ainternet* | U | 25.30 | 13.69 | 70.40 | 84.5 |
| | M | 24.48 | 22.68 | 10.90 | |

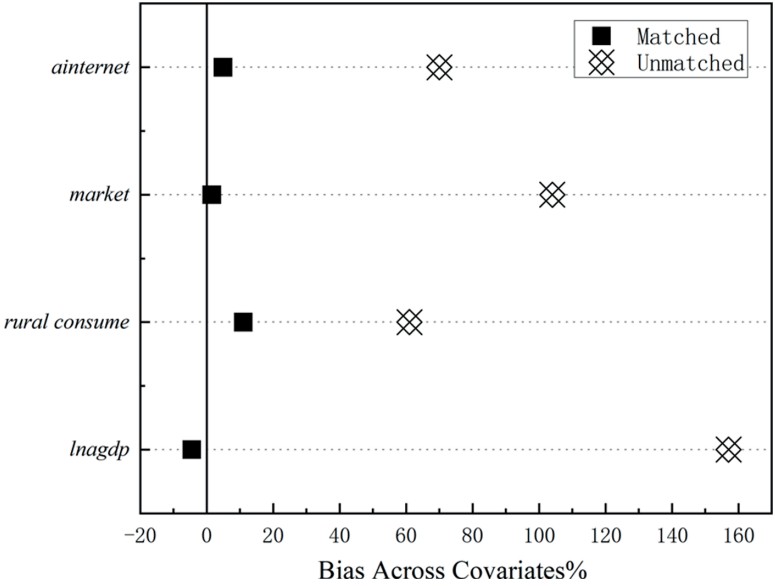

**Fig 3. Standardized deviations.**

### Excluding the impact of other policies

1. Excluding the Impact of Entrepreneurship Policies In 2010, the General Office of the State Council of China issued the Notice on Launching Pilot Entrepreneurship Cities, designating 24 cities, including Beijing, Shanghai, Shenzhen, and Hangzhou, as entrepreneurship pilot cities. The pilot program focused on several key aspects: Improving the Entrepreneurship Service System: Providing training, consultation, guidance, and information for entrepreneurs. Establishing a Venture Investment System: Supporting the development of venture capital funds, business incubators, and entrepreneurship parks. Optimizing the Entrepreneurship Environment: Simplifying registration procedures, reducing taxes and fees for entrepreneurs, and strengthening intellectual property protection. Enhancing Entrepreneurship Talent Development: Encouraging universities, research institutes, and enterprises to cultivate entrepreneurial talent, while supporting high-level and overseas talents in starting businesses. Promoting Entrepreneurship Awareness: Increasing public awareness through campaigns, recognizing exemplary entrepreneurs, and disseminating best practices. Given the close connection between this policy and the subject of this study, a dummy variable based on

the policy's implementation year was constructed for regression analysis to estimate its impact on rural revitalization.

The results are shown in column (3) of Table 6. The findings indicate that the regression coefficient for innovation policy remains positive and significant, while the coefficient for entrepreneurship policy is negative and insignificant. This suggests that entrepreneurship policy has a suppressive effect on rural revitalization. The underlying reasons for this can be explained using the urban migration model [49]. The implementation of entrepreneurship policies may lead to a talent drain from rural areas, reducing human and social capital and thereby impeding rural development. Furthermore, entrepreneurship policies may exacerbate urban-rural disparities, increasing relative poverty and marginalization in rural areas, ultimately hindering rural progress.

2. Excluding the Impact of Smart City Policies Smart cities, as a new paradigm integrating digital technologies with urban development, have been extensively promoted across China since the Ministry of Housing and Urban-Rural Development issued the Notice on Launching National Smart City Pilot Programs in 2012. To date, 290 pilot sites (including districts, counties, and towns) have been established nationwide. These pilots aim to transform cities through the integration of digital technologies, fundamentally altering production and lifestyles.

To examine whether rural revitalization is influenced by this policy, a dummy variable based on the policy's implementation year was constructed for regression analysis to address potential endogeneity issues. The results are presented in column (4) of Table 6. The findings reveal that under the influence of the smart city policy, the regression coefficient for innovation policy remains positive and significant at the 1% level. However, the dummy variable for the smart city policy is negative and only significant at the 10% level. This suggests that the smart city pilot policy may not have fully considered the specific needs of rural revitalization during its design and implementation stages, or that its resource allocation has been more advantageous for urban development rather than rural areas.

**Robustness tests**  To conduct a deeper analysis of how innovation policies drive rural revitalization, this study attempts to replace unstandardized indicators as dependent variables for regression analysis:

**Table 6. Robustness test.**

|  | (1) Cross-sectional PSM | (2) Year-by-year PSM | (3) Excluding the Impact of Entrepreneurship Policies | (4) Excluding the Impact of Smart City Policies |
|---|---|---|---|---|
| Inno_Policy | 0.8143** (2.1893) | *0.5657* (1.8351) | 1.0417*** (2.9853) | 1.0320*** (2.7300) |
| Entrepreneurship | | | -0.1687 (-0.5097) | |
| Smart city | | | | -0.4850* (-1.7062) |
| Control-Var | YES | YES | YES | YES |
| Fixed-effect | YES | YES | YES | YES |
| Observations | 3215 | 3215 | 4,168 | 4,168 |
| $R^2$ | 0.7897 | 0.7040 | 0.7858 | 0.7864 |

1. Rural Per Capita Education Level and Rural Per Capita Income:

   The rural per capita education level reflects the educational attainment of individuals in rural areas, serving as a critical indicator of human capital and the potential for economic development [50,51]. Education can enhance productivity and innovation, thereby contributing to rural revitalization. Accordingly, the average years of education of rural residents is employed as the dependent variable in the benchmark regression analysis.

   The results, as shown in column (2) of Table 7, indicate that the coefficient for the impact of innovation city pilot policies on rural per capita education level is positive and significant at the 5% level. This can be attributed to the fact that innovation city pilot policies often concentrate resources on developing high-tech industries and innovation-driven economies. Economic growth in these cities subsequently drives economic expansion in surrounding areas through capital, technology, and industrial chain effects.Furthermore, innovation cities typically attract top-tier educational resources, including higher education institutions, vocational training centers, and community education programs. The expansion and spillover effects of these resources—through regional collaboration, remote education initiatives, and educational poverty alleviation programs—extend to rural areas, thereby improving local education levels.

   Rural per capita income, as a measure of the average income level of individuals in rural areas, is a key indicator of economic well-being and reflects the level of economic development and prosperity in rural communities [52]. Higher income levels can support investments in infrastructure, services, and other development initiatives, thereby promoting rural revitalization. Accordingly, the average income of rural residents is used as the dependent variable in the benchmark regression analysis. The results, as shown in column (1) of Table 7, indicate that the coefficient of the innovation city pilot policy on rural per capita income is positive and significant at the 1% level. This outcome can be attributed to the policy's adjustment of the education structure to align with the needs of economic and social development [53] and its optimization of educational resource allocation, which helps to narrow urban-rural and regional disparities.

2. Level of Fiscal Support for Agriculture:

   Agriculture serves as the foundation of the national economy, and fiscal support for agriculture significantly enhances the quality of agricultural development by providing funding and resources [54]. This includes supporting agricultural production, improving rural infrastructure, and increasing farmers' income, thereby driving the economic growth of rural areas [55]. Fiscal support for agriculture is a critical instrument for addressing the "three rural issues" (agriculture, rural areas, and farmers). Increased fiscal investment effectively alleviates various challenges faced by rural areas, such as poverty, inadequate education, and poor healthcare, thereby advancing the rural revitalization strategy.

   This study uses the fiscal expenditure on agriculture, forestry, and water affairs in prefecture-level cities as the dependent variable for the benchmark regression analysis. The results, presented in column (3) of Table 7, show that the coefficient of the innovation city pilot policy on fiscal support for agriculture is positive and significant at the 1% level. This is because the policy emphasizes promoting green agricultural development, strengthening policy support, and accelerating the establishment of modern agricultural industrial systems. Such efforts contribute to the sustainable use of agricultural resources and provide long-term and stable funding directions for fiscal support for agriculture. Additionally, the innovation city pilot policy leverages fiscal funds as a guiding mechanism to attract more private capital into the agricultural sector, creating

**Table 7. Robustness tests with alternative indicators.**

| | (1) | (2) | (3) | (4) | (5) |
|---|---|---|---|---|---|
| VARIABLES | Per capita income | Per capita education | Financial agriculture | Integration | Transfer payment |
| *Inno_Policy* | 872.2492** | 0.0694*** | 5.5951*** | 0.0240* | 19.3942*** |
| | (2.5918) | (2.6804) | (2.6344) | (1.8460) | (3.0720) |
| Constant | 29,947.4653*** | 2.2443*** | -82.5443*** | -0.0480 | 23.1782 |
| | (6.1243) | (6.1134) | (-2.7536) | (-0.2522) | (0.2788) |
| Observations | 4,168 | 4,168 | 3,286 | 4,168 | 3,687 |
| R-squared | 0.7853 | 0.7325 | 0.5503 | 0.0269 | 0.4310 |
| Number of city | 278 | 278 | 274 | 278 | 250 |
| CityFE | YES | YES | YES | YES | YES |
| YearFE | YES | YES | YES | YES | YES |

a diversified investment structure that expands financing channels and increases fiscal support for agriculture.

3. Urban-rural integration level: Urban-rural integration helps to eliminate barriers between urban and rural areas, facilitating the free flow of resources such as capital, technology, and talent. This optimized resource allocation brings new development opportunities to rural areas, advancing agricultural modernization and industrial upgrading while narrowing the urban-rural development gap. Urban areas, with their advantages in economics, education, and healthcare, can extend these benefits to rural regions, improving the living standards and well-being of rural residents.

   Drawing on the study by Zhang [56], this research constructs an urban-rural integration index based on five dimensions—population, space, economy, ecology, and society—and uses it as the dependent variable in the benchmark regression analysis. The results, shown in column (4) of Table 7, indicate that the coefficient of the innovation city pilot policy on urban-rural integration is positive and significant at the 10% level. This suggests that the policy optimizes resource allocation, removes barriers between urban and rural areas, and facilitates the free flow of resources, thereby bringing new development opportunities to rural areas and advancing agricultural modernization and industrial upgrading.

4. Transfer Payments: Transfer payments provide essential fiscal resources for rural areas, which can be used for agricultural production, infrastructure construction, and improvements in public services. Agricultural insurance, as a key fiscal support policy for agriculture and farmers in China, is also widely used globally as a tool for managing agricultural risks [57]. Central government fiscal transfer payments provide subsidies for agricultural insurance premiums, and the extent of agricultural insurance implementation in various regions is directly tied to transfer payments [58].

   This study uses the agricultural insurance compensation amount at the prefecture level as the dependent variable for the benchmark regression analysis. The results, shown in column (3) of Table 7, reveal that the coefficient of the innovation city pilot policy on transfer payments is positive and significant at the 1% level. This indicates that the innovation city pilot policy is committed to achieving the goal of common prosperity. Under the policy framework, local governments have increased their focus on underdeveloped regions, allocating funds for development. Additionally, the policy promotes coordinated urban-rural development, gradually narrowing the urban-rural income gap and enabling rural residents to enjoy equal development opportunities and welfare benefits as their urban counterparts.

**Placebo test.** Although this study has examined the control variables, there might still be unobserved factors affecting the results of the innovative policy. To verify the true causal effect of the innovative policy on rural revitalization, this effect should not exist or should be minimal before changing the value of this variable. Based on this principle, this study draws on the research by Adukia and Novosad [59] and utilizes STATA software to construct a dummy variable Inno_Random. This dummy variable is generated completely randomly and is used to replace the core explanatory variable in the original model for a simulated experiment. The model for the simulated experiment is as follows Eq (11):

$$\text{Rural}_{\text{index}\,i,t} = \beta + \beta_1 \text{Inno}_{\text{Random}\,i,t} + \beta_2 \text{Control}_{\text{Var}\,i,t}, + \text{CityFE} + \text{YearFE} + \varepsilon_{i,t} \qquad (11)$$

During the simulation process, multiple random policy generations are applied to all cities in the dataset, followed by regression analysis on the Rural Revitalization Index, recording the values of $\beta_1$. The results are shown in Fig 4. From the graph, it can be observed that after multiple simulations of regression with randomly set experimental groups, the regression coefficients of the core explanatory variable are uniformly distributed around 0. However, the estimated coefficient of the actual policy is 1.296, significantly greater than the placebo test results. This also indicates the robustness of the results, suggesting that the innovative policy studied in this paper indeed drives rural revitalization to a certain extent.

## Heterogeneity analysis

In order to test and analyze the robustness of the causal effects, eliminate potential confounding factors, enhance the credibility of causal inference [60], and assess the varying demands and responses of different city groups to the implementation of innovative policies, thus providing a basis for formulating more effective and equitable policies, heterogeneous analysis is conducted. This mainly includes:

1. The impact of city hierarchy on policy effectiveness:

    Generally, provincial capital cities and municipalities directly under the central government in China exhibit differences in response to science and technology, industrial structure, and innovative policies. Analyzing their heterogeneity helps clarify the effects of city characteristics on policies. Therefore, through heterogeneous analysis, the urban-rural relationship among provincial capitals, municipalities directly under the central government, and ordinary prefecture-level cities can be thoroughly explored, facilitating a deeper understanding of the extent to which cities influence rural areas and the synergistic effects under innovation promotion. Key cities often possess a strong aggregation capacity of innovative elements and leverage their advantages in economic development, policies, and innovation element aggregation to enhance urban innovation competitiveness and levels. This study sets the city hierarchy (Rank) as a dummy variable, with key cities (provincial capitals, municipalities directly under the central government) assigned a value of 1 and ordinary cities assigned a value of 0. A cross-multiplication of the city hierarchy dummy variable and the innovative city pilot policy dummy variable is conducted to obtain $Inno_{Policy} * Rank$, which is then regressed. The results are shown in Table 8 (1).

    From the results, it can be observed that the regression coefficient of the innovative policy variable is significantly positive at the 5% level, while the regression coefficient of the interaction term between the innovative policy and city hierarchy is positive but not significant. This suggests that there is no significant difference in the promoting effect of

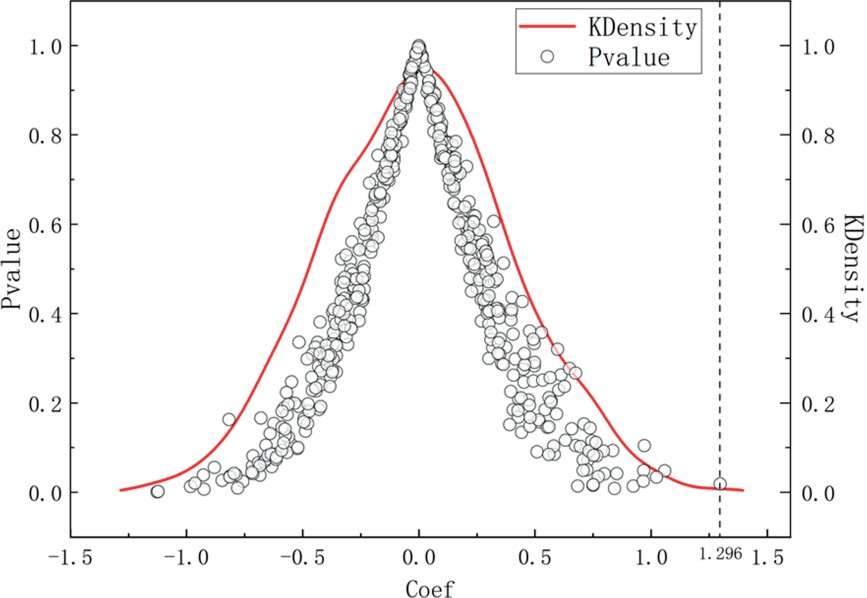

**Fig 4. P-values and Kernel density corresponding to $\beta_1$.**

**Table 8. Heterogeneity analysis.**

| VARIABLES | (1) | (2) | (3) |
|---|---|---|---|
| | **City Hierarchy** | **Educational and Scientific Levels** | **Entrepreneurial Activity** |
| *Inn-Policy* | 0.8690** | 0.9835*** | 0.9706** |
| | (2.5296) | (2.8589) | (2.2643) |
| *Inno-Policy*Rank* | 1.0558 | | |
| | (1.1673) | | |
| *Inno-Policy*Edu-Rec* | | 0.1171 | |
| | | (1.5862) | |
| *Inno-Policy*Entre-Activation* | | | 0.2286** |
| | | | (2.0681) |
| Observations | 2,664 | 2,664 | 2,664 |
| R-squared | 0.7272 | 0.7286 | 0.7266 |
| Number of city | 272 | 272 | 272 |

innovative policies on rural revitalization between key cities and ordinary prefecture-level cities. One possible reason for this phenomenon is that China emphasizes the coordinated development of various innovative elements within cities when implementing innovative city pilot programs, including the rational allocation of resource elements, indirectly driving the simultaneous implementation of policies in key cities and ordinary prefecture-level cities.

2. Heterogeneity in educational and scientific levels:

China's distribution of universities is uneven, with some large cities and developed regions having more and better university resources, while some remote areas have relatively fewer universities [61]. For example, major cities such as Beijing, Shanghai, Nanjing, and Wuhan have multiple prestigious universities, while the number of universities in some western and central-western regions is relatively low. Therefore, drawing from

the research by [62], this study includes university resources as a variable in regression analysis. However, considering that merely measuring prestigious universities cannot accurately assess university resources, in setting up virtual variables for heterogeneous analysis, the specific number of universities (colleges and above) in each city is divided by the local population to obtain $Edu_Rec$. This data is then multiplied by the innovative policy ($Inno_Ppolicy$) to create a new virtual variable, $Inno_Ppolicy * Edu_Rec$, which is then included in the regression. The results are shown in Table 8 (2). From the figures, it is evident that the coefficient of the pilot policy in the regression with virtual variables remains significantly positive at the 1% level. However, while the coefficient of the virtual variable is positive, it is not significant. This indicates that the promotion effect of innovative city pilot policies on rural revitalization does not vary due to differences in the educational and scientific levels of cities. The reason for this phenomenon is that cities lacking educational and scientific resources often require innovation to drive development. Innovative policies can stimulate the vitality of enterprises, research institutions, and individual innovators by providing support and incentives, promoting the development of new technologies and industries.

Additionally, innovative policies can help cities address the issue of inadequate educational and scientific resources by guiding investment and resources towards innovation. By directing resources towards innovation, cities can strengthen the allocation of resources in areas such as scientific research and higher education. Therefore, the rural revitalization effect of innovative policies may be more significant in cities with relatively underdeveloped educational and scientific resources.

3. Heterogeneity in entrepreneurial activity:

Entrepreneurial activity can drive rural economic development, promote job creation, increase income, and improve the living standards of rural residents [63]. Additionally, entrepreneurial activity helps to promote the adjustment and upgrading of rural industrial structure, stimulate diversified rural economic development, and enhance the competitiveness and sustainability of rural economies. Scientifically characterizing urban entrepreneurial activity relies on obtaining the number of new startups in each city. Therefore, this study follows the approach of [64], primarily using the population method in the benchmark regression. This involves using the urban population as the standardized base and the number of new startups per hundred people in the city as the measure of entrepreneurial activity to construct a virtual variable ($Entre_Activation$). Multiplying this variable by the innovative policy ($Inno_Ppolicy$) yields a new virtual variable for regression analysis. The results of the regression are shown in Table 8 (3). The results indicate that the coefficient of $Inno_Ppolicy * Entre_Activation$ is 0.2286, significantly positive at the 5% level. Combined with the average treatment effect of the benchmark regression, which is 0.9835, it can be inferred that innovative policies can drive rural revitalization, with better effects in cities with higher levels of entrepreneurial activity compared to those with lower levels. This is because cities with higher levels of entrepreneurial activity often have richer resources and better environments, including talent, capital, technology, and market access, providing more favorable conditions for innovation. Additionally, based on the Chinese context, cities with higher levels of entrepreneurial activity tend to receive more policy support from the government, including financial subsidies, tax incentives, and technological support, which helps to stimulate innovation vitality.

Based on the heterogeneity analysis conducted earlier using interaction terms, it has been preliminarily determined that the impact of innovation policies on rural revitalization

exhibits certain variations due to differences in city hierarchy, scientific and educational resources, and entrepreneurial activity. To further explore the influence of urban heterogeneity on policy effects, this study conducts a more detailed classification of cities and performs grouped regression analyses according to city categories, specifically including:

1. Central cities:
   In the earlier heterogeneity analysis, preliminary studies were conducted on city hierarchy (municipalities and provincial capitals). However, focusing solely on municipalities and provincial capitals is insufficient, as many cities differ significantly in economic strength, innovation resources, and industrial structures. For example, "new first-tier" cities like Dongguan and Suzhou possess robust R&D capabilities and policy resources, while sub-provincial cities such as Qingdao and Dalian may rely more on regional specialty industries. These disparities in resource endowment may influence the spillover effects of innovation policies on surrounding rural areas. Moreover, variations in government execution capacity, policy design, and the implementation of innovation policies across cities directly impact critical factors such as rural industrial upgrading, infrastructure development, and population return.
   Drawing on the methodology of Zhao and Zhang [65], this study defines municipalities, provincial capitals, sub-provincial cities, and new first-tier cities as "central cities" and divides the sample into two groups: central and non-central cities. Benchmark regressions were conducted separately for each group. Results, as shown in Table 9 (1) and (2), indicate that the coefficient of innovation policies on the rural revitalization index is negative and insignificant for central cities, while for non-central cities, it is positive and significant at the 5% level. This suggests that central cities, already benefiting from abundant resources and policy support, experience lower marginal benefits from innovation pilot policies. In contrast, non-central cities, with weaker infrastructure and resources, derive more direct and significant improvements in rural conditions and industrial development from policy interventions.

2. Coastal cities:
   Due to geographical advantages, coastal cities generally possess more developed economies, attracting domestic and foreign investment and fostering foreign trade and port economies. In comparison, non-coastal cities often face economic constraints, particularly in attracting foreign investment and engaging in international trade. Although some inland cities enhance their economic strength through developing specialized industries and strengthening regional cooperation, their economic performance still lags behind coastal cities [66]. Based on this distinction, the study divides cities into coastal and non-coastal groups for benchmark regression analysis. Results, as shown in Table 9 (3) and (4), reveal that the coefficient of innovation policies on the rural revitalization index is positive and significant at the 5% level in coastal cities, whereas it is positive but insignificant in non-coastal cities. These empirical findings align with theoretical expectations: coastal cities, leveraging their geographical advantages, better facilitate the impact of innovation policies on rural revitalization.

3. Yangtze River cities:
   Cities along the Yangtze River generally enjoy advanced economic development. In particular, the Yangtze River Delta region has established a modernized industrial system [67]. The urban agglomerations in the middle reaches are dominated by industrial and service sectors, while the western regions feature a mix of resource-based and emerging industries. The Yangtze River Basin also exhibits a high degree of regional

**Table 9. Further heterogeneity analysis-1.**

| VARIABLES | (1) | (2) | (3) | (4) |
|---|---|---|---|---|
| | Central Cities | | Coastal Cities | |
| | YES | NO | YES | NO |
| *Inno_Policy* | -1.7056* | 0.7461** | 0.7565** | 0.2872 |
| | (-1.7964) | (2.1021) | (2.0788) | (0.5294) |
| Constant | 88.0733*** | 25.2828*** | 45.1420*** | 8.1588 |
| | (4.0046) | (5.4963) | (7.6750) | (1.2845) |
| Observations | 539 | 3,629 | 1,650 | 1,853 |
| R-squared | 0.7895 | 0.7938 | 0.8809 | 0.7300 |
| Number of city | 36 | 242 | 110 | 168 |
| CityFE | YES | YES | YES | YES |
| YearFE | YES | YES | YES | YES |

economic integration, with distinct east-to-west gradients and significant collaborative effects in industrial division of labor.

Based on the Yangtze River Economic Belt Development Plan Outline issued by the Chinese government in 2016, this study categorizes cities into Yangtze River cities and non-Yangtze River cities for benchmark regression analysis. The results, shown in Table 10 (5) and (6), indicate that the coefficient of innovation policies on the rural revitalization index is positive and significant at the 1% level for Yangtze River cities, whereas for non-Yangtze River cities, the coefficient is positive but not statistically significant.

The empirical findings align with the theoretical analysis. Yangtze River cities benefit from their unique geographical location and the Chinese government's strong emphasis on the development of the Yangtze River Economic Belt, supported by several strategic policy documents, such as the Yangtze River Economic Belt Development Plan Outline. These policies not only promote the coordinated development of urban clusters but also emphasize rural revitalization and urban-rural integration. As a result, innovation policies in Yangtze River cities often receive greater fiscal support and exhibit stronger implementation efficiency, effectively driving rural industrial development and ecological improvement. This amplifies the impact of innovation policies on rural revitalization.

4. Population:

High population density often indicates greater market demand, abundant labor resources, and stronger consumption power, all of which are critical drivers of economic growth [68]. In the context of rural revitalization, densely populated areas are more likely to achieve economies of scale, promote industrial agglomeration, and foster specialized division of labor, thereby improving productivity and economic efficiency. Conversely, less densely populated areas may better implement sustainable development strategies, as reduced population pressure facilitates ecological preservation and rational resource utilization.

To examine population-based heterogeneity among cities, this study uses the Notice on Adjusting the Urban Scale Classification Standards (State Council Document No. 51, 2014) issued by the State Council of China. Cities are classified into large cities and medium-to-small cities based on a population threshold of one million residents within municipal districts. Benchmark regressions were conducted separately. Results, as shown in Table 10 (7) and (8), indicate that the coefficient of innovation policies on the rural revitalization index is positive and significant at the 5% level in large cities,

**Table 10. Further heterogeneity analysis-1.**

| VARIABLES | (5) | (6) | (7) | (8) |
|---|---|---|---|---|
| | Yangtze River | | By Population | |
| | YES | NO | Large Cities | Small and Medium-sized Cities |
| *Inno_Policy* | 1.1624*** | 0.8194 | 0.9340** | 0.7530 |
| | (3.3252) | (1.3657) | (2.1275) | (1.0166) |
| Constant | 15.6728* | 40.4695*** | 50.0507*** | 27.7857*** |
| | (1.8890) | (7.0407) | (5.1074) | (5.0839) |
| Observations | 1,544 | 2,624 | 1,303 | 2,820 |
| R-squared | 0.7884 | 0.7906 | 0.8235 | 0.7978 |
| Number of city | 103 | 175 | 87 | 191 |
| CityFE | YES | YES | YES | YES |
| YearFE | YES | YES | YES | YES |

whereas it is positive but insignificant in small cities. This finding can be explained as follows: larger cities, with higher labor demand from rural areas, provide more employment opportunities and facilitate the flow of rural products into urban markets. Furthermore, larger cities feature more complex social networks, fostering stronger cultural exchange and mutual recognition between rural and urban areas, which enhances resource complementarities.

## Mechanism testing

The intermediary effect is crucial in economic research, helping us understand the mechanism through which independent variables affect dependent variables and revealing the pathways of interaction between variables [69]. Through intermediary effects, we can analyze research results more comprehensively and quantify the intermediate processes in causal relationships. Therefore, in order to innovate the mechanism of the effect of policy on rural revitalization, this paper borrows from the analytical approach of the intermediary effect model [70], constructing an intermediary effect model as shown in Eqs (12), (13, and (15).

$$M_{i,t} = \alpha + \alpha_1 Inno\_Policy_{i,t} + \alpha_2 Control\_Var_{i,t} + CityFE + YearFE + \varepsilon_{i,t} \quad (12)$$

$$Rura\_index_{i,t} = \gamma + \gamma_1 M_{i,t} + \gamma_2 Inno\_Policy_{i,t} + \gamma_3 Control\_Var_{i,t} \quad (13)$$

$$+ CityFE + YearFE + \varepsilon_{i,t} \quad (14)$$

$$Contribute_i = \frac{\alpha_1 * \gamma_2}{\alpha_1 * \gamma_2 + \gamma_1} \quad (15)$$

$M_{i,t}$ represents the intermediary variable, *Contribute_i* indicates the contribution rate of the intermediary effect to the total effect, and the remaining parameters are explained as in Eq (1). Simultaneously, the analysis will be conducted from the following four directions:

(1) Mediating effect of communication development level: The development of communication technology strengthens the connection and communication between urban and rural areas, promoting the sharing and flow of resources. The digital interconnection between rural and urban areas helps to drive integrated industrial development. The coverage and application of communication facilities also enable rural residents to access more education, medical, and cultural public services.Kenny [71] believe that

the internet has great potential in reducing poverty and promoting sustainable development in rural areas. Therefore, this study uses the proportion of administrative villages with Internet access (Correspondence) to reflect the local level of communication development.

(2) Mediating effect of cultural education: Nachtigal and Director [72] argue that culture shapes the values, beliefs, and spirit of exploration in rural areas, providing the foundation for social cohesion and identity. It helps to preserve traditional knowledge and practices, contributing to sustainable development and the well-being of rural communities. Education promotes critical thinking, problem-solving, and innovation, which are crucial for addressing the challenges faced by rural communities and promoting sustainable development. Therefore, this study selects the proportion of local residents' cultural education expenditure to total expenditure (Culture) to reflect the local level of cultural education.

(3) Mediating effect of policy aggregation: To measure the effect of policy aggregation, this study uses big data analysis methods based on the Peking University Law Database to collect the number of innovative policies issued by 280 sample cities from 2004 to 2018. Using "city name" and "innovation" as keywords for retrieval, the number of policy documents issued by municipal and higher administrative units other than pilot policies (Policy) is recorded to measure the support effect of policies.

(4) Mediating effect of talent absorption: The development of rural China relies on talent support. Shi and Lai [73] believe that the number of talents with high academic qualifications reflects the innovation level of enterprises and is conducive to their development. However, considering the specificity of rural development, there are hardly any high-tech enterprises in the initial stage of development. Therefore, this study uses the proportion of undergraduate and above teachers in rural compulsory education stage (Degree) to characterize the local talent level.

Regression analysis of the above intermediary effect models will be conducted, and the significance of the mechanism will be tested based on the Bootstrap method, as shown in Table and Table 11 (1) and (2) respectively show the regression results of innovative policies on local communication development and the regression results of communication development on rural revitalization level. Regression (1) indicates that the regression coefficient of innovative policies on rural communication level is significantly positive at the 5% level, indicating that innovative pilot policies have a certain strengthening effect on local communication development. Result (2) shows that the regression coefficient of communication development on rural revitalization is significantly positive at the 1% level. Overall, the communication development level passes the Bootstrap test, with a contribution rate of 52%, indicating that innovative policies promote rural communication development, thereby promoting rural revitalization, and the mediating effect of communication development is established.

Table 11 (3) and (4) show the regression results of innovative policies on local cultural education development and the regression results of cultural education development on rural revitalization, respectively. The regression result (3) shows that the regression coefficient of innovative policies on cultural education development is significantly positive at the 5% level, indicating that innovative policies guide local residents to pay more attention to cultural education. Regression result (3) shows that the regression result of local cultural education development on rural revitalization is significantly positive at the 1% level. Overall, the cultural education level passes the Bootstrap test, with the mediating effect contributing 45% to the total effect, indicating that innovative policies promote local cultural education level,

**Table 11. Regression results of mediating effects.**

| VARIABLES | (1) Correspondence | (2) Rural_Index | (3) Culture | (4) Rural_Index |
|---|---|---|---|---|
| *Inno_Policy* | 0.5967** | 0.6208*** | 0.5255** | 0.7007*** |
| | (2.4242) | (2.8492) | (2.1699) | (2.9284) |
| *Correspondence* | | 1.1317*** | | |
| | | (36.7855) | | |
| *Culture* | | | | 1.1333*** |
| | | | | (34.4040) |
| Bootstrap | 2.953 Z=4.31 | P=0 | 2.752 Z=3.86 | P=0 |
| Contribution Rate | 52 | | 45 | |
| Observations | 2,664 | 2,664 | 2,664 | 2,664 |
| R-squared | 0.5709 | 0.8672 | 0.5857 | 0.8611 |
| Number of city | 272 | 272 | 272 | 272 |
| CityFE | YES | YES | YES | YES |
| YearFE | YES | YES | YES | YES |

thereby promoting rural revitalization, and the mediating effect of cultural education level is established.

Table 12 (5) and (6) respectively show the regression results of innovative policies on policy aggregation effect and the regression results of policy aggregation response on rural revitalization. The regression results show that the regression coefficients of innovative policies on policy aggregation effect and policy aggregation effect on rural revitalization are all significantly positive. Meanwhile, the policy aggregation effect passes the Bootstrap test, with the mediating effect contributing 9% to the total effect. This indicates that innovative policies strengthen local policy aggregation, thereby promoting rural revitalization, and the mediating effect of policy aggregation is established. Table 12 (7) and (8) respectively show the regression results of innovative policies on talent aggregation and the regression results of talent aggregation on rural revitalization. The regression result (7) shows that the regression coefficient of innovative policies on talent aggregation is significantly positive at the 5% level, indicating that innovative policies strengthen local talent level. Regression result (8) shows that the regression coefficient of talent aggregation effect on rural revitalization is significantly positive, indicating that talent aggregation enhances local productivity, thereby promoting rural revitalization. Overall, the talent aggregation effect passes the Bootstrap test, with a contribution rate of 62.5% to the total contribution rate, indicating that innovative policies drive local talent aggregation effect, thereby promoting rural revitalization, and the mediating effect is established.

From the above analysis, it is evident that although the levels of communication, cultural education development, policy aggregation, and talent aggregation all have significant positive effects on the level of rural revitalization, talent aggregation contributes the most, while policy aggregation contributes the least. This suggests that in future development processes, apart from fully leveraging the talent aggregation effect driven by innovation-driven policies to enhance rural development through talent aggregation, there is also a need to further strengthen the guiding role of innovation policies in the implementation of other policies, broaden the limitations of policy implementation, and better promote the level of rural revitalization.

**Table 12. Regression results of mediating effects-2.**

| VARIABLES | (5)<br>Policy | (6)<br>Rural_Index | (7)<br>Degree | (8)<br>Rural_Index |
|---|---|---|---|---|
| *Inno_Policy* | 3.5800***<br>(3.3402) | 1.1688***<br>(2.8177) | 0.6969***<br>(2.8708) | 0.4846**<br>(2.2275) |
| *Policy* | | 0.0356**<br>(2.1722) | | |
| *Degree* | | | | 1.1646***<br>(35.8024) |
| Bootstrap | 0.8308 Z=3.17 | P=0.002 | 3.0626 Z=4.23 | P=0 |
| Contribution Rate | 9.4 | | 62.5 | |
| Observations | 2,664 | 2,664 | 2,664 | 2,664 |
| R-squared | 0.2807 | 0.7279 | 0.5901 | 0.8653 |
| Number of city | 272 | 272 | 272 | 272 |
| CityFE | YES | YES | YES | YES |
| YearFE | YES | YES | YES | YES |

## Conclusions and policy implications

### Main contributions of this study

Drawing on data from 280 prefecture-level cities between 2004 and 2018, the study employs a multi-period DID (Difference-in-Differences) model to systematically assess the effects of innovation-driven policies on rural revitalization. The results demonstrate that these policies have significantly improved rural revitalization in pilot areas during the study period, with findings robust across various sensitivity tests. The study also investigates heterogeneity in policy effects, focusing on city tier, scientific and educational resources, and entrepreneurial activity. Subgroup analyses reveal that regions with higher entrepreneurial activity experience stronger policy effects, while city tier and resource availability have limited influence. Additionally, non-central cities, coastal regions, Yangtze River basin areas, and populous cities benefit more significantly from the policy.

Mechanism testing identifies key pathways through which innovation policies impact rural revitalization, including the development of the telecommunications industry, advancements in education and culture, talent accumulation, and policy clustering effects. Among these, talent accumulation emerges as the most influential factor, while policy clustering contributes relatively less. These findings highlight the multifaceted ways innovation policies drive rural development, providing theoretical insights and practical implications for fostering urban-rural integration and advancing rural modernization.

This study provides three significant contributions. First, it evaluates the impact of innovation-driven policies on rural revitalization in China by employing a quasi-natural experiment based on national innovation-driven city pilot policies. This approach not only broadens the scope of research on innovation-driven policies but also enriches rural revitalization studies within the context of China's policy landscape. Second, it optimizes the evaluation index system for rural revitalization, allowing for more accurate assessments at the provincial level. By examining heterogeneous effects across ownership types and innovation inputs, the study uncovers the complexities of China's innovation policy framework, offering valuable insights for both domestic and global researchers. Third, it constructs a comprehensive econometric framework that integrates multiple years and city-level innovation policies, providing a deeper understanding of the mechanisms driving innovation-led rural revitalization. The analysis also explores heterogeneous effects by examining administrative levels,

scientific and educational resources, and entrepreneurial activities, offering theoretical and practical guidance for integrating innovation and rural development.

## Policy implications of this study

1. Strengthen the implementation of innovation policies to fully leverage their effects on rural revitalization. This study shows that innovation policies have a significant effect on promoting rural revitalization. As a policy arrangement aimed at facilitating the transition between old and new growth drivers, innovation policies effectively enhance the level of rural revitalization. The successful experiences of pilot cities can provide references for other regions. By summarizing the successful paths of pilot cities, actionable policy guidance documents can be developed to support the promotion of these experiences in other cities. Through a "pilot first, then expand" approach, pilot experiences can gradually be extended to more areas, particularly rural areas with weaker resource endowments. Pilot cities are encouraged to formulate "innovation spillover" plans, such as providing special funds to support rural technology transfer or the implementation of innovative projects. Regional innovation hubs can be built, forming regional innovation networks centered on pilot cities, to drive broader rural development.

2. Emphasize the role of rural revitalization policies in communication, talent, culture and education, and policy support, and expand policy transmission channels. Policy formulation should focus on expanding the coverage of high-speed broadband, 5G networks, and the Internet of Things (IoT) in rural areas to provide technical support for smart agriculture, e-commerce, and online education. The implementation of the Information for All project should be promoted, reducing the barriers for rural residents to access information through communication technology, and ensuring that innovation policies and resources reach rural areas quickly. Encourage rural areas to explore and innovate cultural resources, develop cultural products, rural tourism, and intangible cultural heritage protection. Use digital technology to promote rural culture to broader markets, enhancing its influence and economic benefits. Encourage the flow of urban educational resources to rural areas through policies, such as establishing paired educational support programs or remote education platforms. Provide specialized training in modern agricultural technologies, e-commerce, and entrepreneurship for rural youth and farmer entrepreneurs to enhance their overall capabilities. Strengthen collaboration between agriculture, education, communication, and cultural departments to develop comprehensive rural revitalization support programs. Rural areas should take a leading role in policy implementation to ensure that resources truly benefit grassroots communities.

3. Refine innovation policies to be tailored to local conditions, enhance policy targeting, and avoid a "one-size-fits-all" approach. The findings of this study indicate that while the effects of rural revitalization policies are not influenced by differences along the Yangtze River region, variations still exist across regions with different administrative levels and geographical locations. Therefore, in policy formulation, it is essential to first design targeted policies based on the functional positioning, resource endowments, and development stages of cities and rural areas. For central cities, the focus should be on encouraging the support of surrounding rural development through technological spillover effects and avoiding the "retention" of resources within cities. Comprehensive policies should be developed that integrate innovation policies with goals such as urban-rural integration and equalization of public services. For non-central cities, the direct driving force of policies on rural development should be further strengthened.

Secondly, policy formulation should consider the resource endowments and development bottlenecks of non-coastal cities and design targeted innovation policies. For example, priority should be given to supporting agricultural modernization, the integration of culture and tourism, and the development of regional specialty industries. Greater financial support or technology transfer policies should be provided to non-coastal cities to reduce their resource acquisition gap with coastal cities. Thirdly, for large cities, strengthening mechanisms for technological spillover is crucial, while small and medium-sized cities should encourage high-level innovation talents and teams to settle in these cities by offering benefits such as housing subsidies and research funding. Large cities should be encouraged to collaborate with surrounding small and medium-sized cities to build industrial chains and promote the decentralization of innovation resources. For example, large cities can provide technical support and market channels, while small and medium-sized cities offer industrial support and land resources. Finally, differentiated policy designs should be emphasized for Yangtze River region cities to leverage the advantages of the Yangtze Economic Belt, such as strengthening upstream and downstream industrial collaboration and developing a green ecological economy. Establish a dynamic monitoring mechanism for policy impacts, regularly evaluate the actual effects of policies in different regions, and adjust policy directions based on evaluation results.

## Limitations and future research directions

This study primarily examines the effects of innovation-driven policies on rural revitalization from the perspective of the rural revitalization index at the prefecture-level city scale. However, due to data availability, the data used in this analysis are limited to the prefecture-level administrative areas where the towns (or townships) covered by various innovation city pilot policies are located. Since China's rural development paths have varied significantly since the reform and opening-up era—for instance, villages such as Huaxi and Changjiang within Jiangyin City, Jiangsu Province, exhibit distinct trajectories—future research should focus on refining data to the township level. Doing so would more accurately reflect the performance and multi-path configurations of innovation policies in promoting rural revitalization. This remains a key area for future research.

## Supporting information

**S1 Data. Data in the paper.** We have uploaded the data for the article to DRYAD. The data link is: DOI: 10.5061/dryad.qv9s4mwr5. For detailed data explanations, please refer to the README in the data link.
(XLS)

## Author contributions

**Conceptualization:** Zhige Ying, Zhe Wu.

**Data curation:** Cheng Zhang.

**Project administration:** Hande Chen.

**Software:** Zhiyuan Cheng.

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
