## [Decision Letter · Decision Letter 0]

12 Nov 2024

PONE-D-24-41797Does innovation drive rural revitalization in China? - An empirical study based on Chinese rural areasPLOS ONE

Dear Dr. Ying,

Thank you for submitting your manuscript to PLOS ONE. After careful consideration, we feel that it has merit but does not fully meet PLOS ONE’s publication criteria as it currently stands. Therefore, we invite you to submit a revised version of the manuscript that addresses the points raised during the review process.

In particular, the reviewers point to shortcomings in the literature search and recommend that the framework of the theoretical analysis in terms of methods should be developed prior to the empirical analysis. Also, the method of calculating the rural revitalisation index is not clear and the section of the empirical analysis lacks descriptive statistics of the main variables, i.e. the text does not similarly detail which cities were included in the National Innovation City in which period. They also mention that the reference is somewhat outdated, with fewer references in the last three years.

We look forward to receiving your revised manuscript.

Kind regards,

Václav Venkrbec, Ph.D.

Academic Editor

PLOS ONE

Journal Requirements:

2. Thank you for stating the following financial disclosure: [Natural Science Foundation of Hubei Province(2017AAA408)].

Please include this amended Role of Funder statement in your cover letter; we will change the online submission form on your behalf."

Reviewers' comments:

Reviewer's Responses to Questions

**Comments to the Author**

1. Is the manuscript technically sound, and do the data support the conclusions?

Reviewer #1: Yes

Reviewer #2: Yes

2. Has the statistical analysis been performed appropriately and rigorously? 

Reviewer #1: Yes

Reviewer #2: Yes

3. Have the authors made all data underlying the findings in their manuscript fully available?

Reviewer #1: Yes

Reviewer #2: Yes

4. Is the manuscript presented in an intelligible fashion and written in standard English?

Reviewer #1: Yes

Reviewer #2: Yes

5. Review Comments to the Author

Reviewer #1: Rural revitalization needs the support of innovation policy, and improving innovation capacity is also an important aspect of rural revitalization. The paper has a good value of topic selection, the empirical analysis is more comprehensive, and the overall conclusions are more reliable, but there are still some problems, and this paper suggests a major revision.The review comments are in the annex

Reviewer #2: The article has clear research question and well organized structure. Innovative research conclusions were drawn from a normative empirical analysis based on sufficient data amount and reasonable methodology. There still are some problems that need further modification or interpretation.

6. PLOS authors have the option to publish the peer review history of their article (what does this mean?). If published, this will include your full peer review and any attached files.

Reviewer #1: No

Reviewer #2: No

---

## [Author Response · Author response to Decision Letter 1]

24 Nov 2024

Dear Editors and Reviewers:

Thank you for your letter and for the reviewers’ comments concerning our manuscript entitled “How Does Innovation Policy Drive Rural Revitalization in China? - An Empirical Study Based on Rural China”.Those comments are all valuable and very helpful for revising and improving our paper, as well as the important guiding significance to our researches. We have studied comments carefully and have made correction which we hope meet with approval. In the revised version, we have highlighted all the changes in yellow.The specific revisions can be found in the response letter to the reviewers.

---

## [Decision Letter · Decision Letter 1]

10 Dec 2024

PONE-D-24-41797R1How Does Innovation Policy Drive Rural Revitalization in China? - An Empirical Study Based on Rural ChinaPLOS ONE

Dear Dr. Ying,

Thank you for submitting your manuscript to PLOS ONE. After careful consideration, we feel that it has merit but does not fully meet PLOS ONE’s publication criteria as it currently stands. Therefore, we invite you to submit a revised version of the manuscript that addresses the points raised during the review process.

We look forward to receiving your revised manuscript.

Kind regards,

Václav Venkrbec, Ph.D.

Academic Editor

PLOS ONE

Journal Requirements:

Additional Editor Comments:

The reviewers commend the substantial improvements made to the manuscript following revisions, noting its enhanced structure, writing quality, and adherence to academic standards. Reviewer 1 finds the paper well-prepared and ready for acceptance. Reviewer 2 acknowledges the advancements but highlights areas for minor revision, particularly in standardizing formatting, refining terminology, and clarifying the theoretical framework. Recommendations include correcting missing or unclear section numbering, revising table formatting, and expanding the introduction to include broader international perspectives on innovation and rural revitalization. Furthermore, a distinct chapter on the theoretical mechanism is suggested to strengthen the framework and hypotheses. These minor revisions will solidify the manuscript’s clarity and scholarly impact.

Reviewers' comments:

Reviewer's Responses to Questions

**Comments to the Author**

1. If the authors have adequately addressed your comments raised in a previous round of review and you feel that this manuscript is now acceptable for publication, you may indicate that here to bypass the “Comments to the Author” section, enter your conflict of interest statement in the “Confidential to Editor” section, and submit your "Accept" recommendation.

Reviewer #1: All comments have been addressed

Reviewer #2: All comments have been addressed

2. Is the manuscript technically sound, and do the data support the conclusions?

Reviewer #1: Yes

Reviewer #2: Yes

3. Has the statistical analysis been performed appropriately and rigorously? 

Reviewer #1: Yes

Reviewer #2: Yes

4. Have the authors made all data underlying the findings in their manuscript fully available?

Reviewer #1: Yes

Reviewer #2: Yes

5. Is the manuscript presented in an intelligible fashion and written in standard English?

Reviewer #1: Yes

Reviewer #2: Yes

6. Review Comments to the Author

Reviewer #1: The author has provided excellent responses to all the revision suggestions, and the quality of the revised article has been greatly improved. I have no further questions about this article, which is a good study.

Reviewer #2: The structure, English writing and standardization of the article have been greatly improved in the revision, but the following weak links should be enhanced, mainly in the introduction, theoretical mechanism and standardization of writing.

7. PLOS authors have the option to publish the peer review history of their article (what does this mean?). If published, this will include your full peer review and any attached files.

Reviewer #1: No

Reviewer #2: No

---

## [Author Response · Author response to Decision Letter 2]

18 Dec 2024

Dear Editor and Reviewer,

We feel great thanks for your professional review work on our article. As you are concerned, there are several problems that need to be addressed. According to your nice suggestions, we have made extensive corrections to our previous draft, The details can be found in the attachment.

---

## [Decision Letter · Decision Letter 2]

10 Jan 2025

Does innovation policy drive rural revitalization in China? - An empirical study based on Chinese rural areas

PONE-D-24-41797R2

Dear Dr. zhige Ying

We’re pleased to inform you that your manuscript has been judged scientifically suitable for publication and will be formally accepted for publication once it meets all outstanding technical requirements.

Kind regards,

Basil Msuha

Guest Editor

PLOS ONE

Additional Editor Comments (optional):

Reviewers' comments:

Reviewer's Responses to Questions

**Comments to the Author**

1. If the authors have adequately addressed your comments raised in a previous round of review and you feel that this manuscript is now acceptable for publication, you may indicate that here to bypass the “Comments to the Author” section, enter your conflict of interest statement in the “Confidential to Editor” section, and submit your "Accept" recommendation.

Reviewer #1: All comments have been addressed

Reviewer #2: All comments have been addressed

2. Is the manuscript technically sound, and do the data support the conclusions?

Reviewer #1: Yes

Reviewer #2: Yes

3. Has the statistical analysis been performed appropriately and rigorously? 

Reviewer #1: Yes

Reviewer #2: Yes

4. Have the authors made all data underlying the findings in their manuscript fully available?

Reviewer #1: Yes

Reviewer #2: Yes

5. Is the manuscript presented in an intelligible fashion and written in standard English?

Reviewer #1: Yes

Reviewer #2: Yes

6. Review Comments to the Author

Reviewer #1: The author has provided excellent responses to all the suggested revisions, and the quality of the article has been further improved. I have no further questions about this article, which is a great study.

Reviewer #2: All my comments have been revised or explained in this revesion. The article has sufficient empirical analysis, and the theoretical analysis has been strengthened in the revision. The existing problems are that the literature review is a little bit insufficient, and the conclusion and discussion parts are worth further exploration. I personally think the article basically meet the requirement for publication. Therefore, I have no opinion on revising the article any more. My review is completed.

7. PLOS authors have the option to publish the peer review history of their article (what does this mean?). If published, this will include your full peer review and any attached files.

Reviewer #1: No

Reviewer #2: No

---

## [Editor Report · Acceptance letter]

PONE-D-24-41797R2

PLOS ONE

Dear Dr. Ying,

I'm pleased to inform you that your manuscript has been deemed suitable for publication in PLOS ONE. Congratulations! Your manuscript is now being handed over to our production team.

Kind regards,

on behalf of

Dr. Basil Msuha

Guest Editor

PLOS ONE